# Korean Translation and Psychometric Evaluation of Korean Version EORTC QLQ-BRECON23

**DOI:** 10.3390/ijerph17249163

**Published:** 2020-12-08

**Authors:** Soo-Kyung Bok, Youngshin Song, Ancho Lim, Hyunsuk Choi, Hyunkyung Shin, Sohyun Jin

**Affiliations:** 1College of Medicine, Chungnam National University, Daejeon 35015, Korea; skbok111@gmail.com; 2College of Nursing, Chungnam National University, Daejeon 35015, Korea; ilsu77@hanmail.net (H.C.); allpm1111@o.cnu.ac.kr (H.S.); iyoyoyo0128@naver.com (S.J.)

**Keywords:** breast cancer, reconstruction, validation, quality of life, factor analysis

## Abstract

The purpose of this study was to evaluate the psychometric properties of the Korean version of the European Organization for Research and Treatment of Cancer Quality of Life-QLQ-BRECON23 in women diagnosed and treated for breast cancer undergoing all types of breast reconstruction. Methods: A total of 148 Korean women who underwent breast reconstruction were recruited from the breast cancer center to participate in the study. After performing forward and backward translation of the original English version of the questionnaire into Korean, its validity (construct, known-group validity, concurrent) and reliability were assessed. A structural equation model (SEM) was used to assess construct validity. Results: The mean age of the patients was 52 years, and 89.8% underwent implant-based reconstruction. Construct validity using confirmatory factor analysis showed a good fit, and the effect size was small-to-medium regarding known-group validity. Concurrent validity was confirmed by the significant correlation between the QLQ-BRECON23 and the QLQ-BR23. The reliability of the QLQ-BRECON23 symptom and function scales ranged from 0.61 to 0.87. Conclusion: The Korean QLQ-BRECON23 can be applied to assess quality of life and its related factors, and also to internationally compare the level of quality of life in breast cancer patients undergoing breast reconstruction.

## 1. Introduction

The age distribution of breast cancer cases in South Korea is 12.7% for those in their 30s and 37.1% for those in their 40s, which is lower than in Western countries [1]. With the increasing incidence of breast cancer among young women, BR is also on the rise. For example, there were 99 BR cases in 2000 in Korea but 910 in 2012, an almost tenfold increase [2]. An increase in this rate is expected given that the National Healthcare Insurance System (NHIS) has covered BR surgery since 2015 [2].

Breast reconstruction (BR) after mastectomy, including autologous tissue (flap) reconstruction, implant reconstruction, or a combination of both, has been increasing with the development of breast cancer management worldwide [3]. From the patients’ point of view, BR can affect not only their physical shape but also their psychosocial well-being and daily functioning [4]. With the increased breast cancer survival rate and progress in treatment strategies, the health-related quality of life (QoL) of breast cancer survivors has an important impact on their treatment methods and even surgical procedures [5]. In particular, it has been reported that BR after a mastectomy has a positive effect on women’s body image and their health-related QoL [5,6].

The European Organization for Research and Treatment of Cancer (EORTC) developed a questionnaire to measure the health-related QoL for cancer patients, which has been extensively used worldwide [7]. Meanwhile, in order to assess the health-related QoL of breast cancer survivors, a breast-cancer-specific module, namely the EORTC QLQ-BR23 questionnaire, has been used regardless of whether women have BR surgery or not [7]. Generally, the timing and operation type for BR are chosen according to patients’ preference, physical condition, and risk factors, and its outcomes are also varied [3]. Depending on the type of BR, complications such as capsular contracture, infection, rupture, necrosis of skin flap, flap shrinkage, and reconstructive failure should be considered and monitored [3,4]. However, the EORTC QLQ-BR23 questionnaire does not reflect the particular situation of women undergoing BR [7]. To overcome this limitation, the EORTC developed the QLQ-BRECON23 for measuring QoL in women undergoing BR [8], and the validity and reliability of the original English version were confirmed through an international validation process including seven countries [9].

The QLQ-BRECON23 questionnaire has a total of 23 items: 6 items related to symptoms and 17 items related to functional aspects. The QLQ-BRECON23 questionnaire measures QoL in relation to treatment side effects symptoms, donor site symptoms, loss of nipple, status regarding sexual functioning, satisfaction with breast cosmetic, satisfaction with nipple cosmetic, satisfaction with surgery, satisfaction with donor scars, and preservation/reconstruction of nipple [9]. To assess the psychometric properties of the English version of the QLQ-BRECON23 questionnaire, multi-trait scaling (item-total correlation), content validity, construct validity, convergent/discriminant validity, known-group comparison, and Cronbach’s alpha (reliability) were investigated, and the findings indicated a valid and reliable tool to measure the QoL of women undergoing BR [9]. In particular, the QLQ-BRECON23 questionnaire was significantly correlated with existing scales such as the QLQ-BR23 [9].

Until now, the QLQ-BRECON23 questionnaire has been widely used in English; however, there is no evidence yet on the use of its Korean version. Therefore, the aims of this study were to address the need for a Korean version of the QLQ-BRECON23 by performing a standard translation process of the scale into Korean and to evaluate its psychometric properties with a sample of women undergoing BR to confirm its cultural adaptation in South Korea.

## 2. Materials and Methods

### 2.1. Study Design

A cross-sectional descriptive study design was conducted to examine the psychometric properties of the Korean version of the QLQ-BRECON23 in women undergoing BR.

### 2.2. Participants

Participants were recruited from the breast cancer center of C-National University Hospital in South Korea between February and April 2020. Patients were eligible if they were women above 18 years of age who had undergone reconstructive surgery after being diagnosed with breast cancer. Written consent was obtained from all participants in the study. Exclusion criteria included lack of consent to participate in the study and inability to understand or complete the questionnaires. For this reason, the number of ineligible patients of this reason was two international women who did not understand Korean. There were no restrictions related to cancer stage, type of surgery, or postoperative treatment. Initially, eligible participants were 169 women, but 21 declined to participate (refused 14, depressive mental problem 3, and poor general condition 4). Thus, a total of 148 participants finally completed the questionnaire.

The sample size for confirmatory factor analysis (CFA) can be calculated as 5 to 10 cases per item [10]; thus, at least 115 to 230 participants were needed to confirm the construct of the QLQ-BRECON23 (23 items). Finally, this study used a sample size of 148. 

### 2.3. Procedure

To develop the Korean version of the QLQ-BRECON23 and evaluate its psychometric properties in Korea, two steps were performed. First, the standard translation procedure by the EORTC QoL group was applied [11] as follows: forward translation by two translators, reconciled translation by a third reviewer, and back-translation with comments by two different translators. Subsequently, the preliminary Korean version underwent pilot testing with a sample of 10 target patients to assess difficulties and confusing wording. However, there were no questions or revisions for the preliminary Korean version. The final Korean version of QLQ-BRECON23 was confirmed by the EORTC QoL group.

Second, the psychometric properties of the Korean version of the QLQ-BRECON23 were evaluated, namely content validity, construct validity, known-group validity, concurrent validity, and reliability. Eight healthcare providers checked the items to confirm the content validity. Item analysis and CFA were used to confirm construct validity for multiple items. CFA involves inferential statistics that allow for hypothesis testing about a set of measures, which can lead to a more objective interpretation of dimensionality than exploratory factor analysis [12]. Known-group validity was evaluated in terms of single-item comparisons between groups according to clinical features. Moreover, the correlation between QLQ-BRECON23 and QLQ-BR23 was tested to assess concurrent validity. Lastly, reliability was assessed in terms of Cronbach’s alpha for the symptom and function scales.

### 2.4. Measures

*QLQ-BRECON23*: Self-reported health-related QoL was assessed for symptoms (6 items) and functions (17 items) among patients who underwent BR using the QLQ-BRECON23. In order to measure the health-related QoL for women who underwent post-mastectomy BR, all 23 items are used, whereas a 14-item version can be applicable to women before undergoing mastectomy and BR [9]. The questionnaire contains symptom scales (treatment side effects, TS; donor site symptoms, DS; loss of nipple, NL) and function scales (sexual function with breast, SX, satisfaction with breast cosmetic, SBC, satisfaction with nipple cosmetic, SNC; satisfaction with surgery, SSU; satisfaction with donor scars, SDS; preserve/reconstruct nipple, NP), making 6 multi-item scales (TS, DS, SX, SBC, SNC, SSU), and 3 stand-alone items (NL, SDS, NP). The SNC, DS, NL, SDS, and NP scales are conditional and can only be scored if applicable to the patient.

All of the scales and single-item measures range in score from 0 to 100; a high score on the symptom scales and single items represents a high level of symptomatology or problems, whereas a high score on the functional scales and single items represents a high level of functioning or satisfaction. Cronbach’s alpha ranged from 0.67 to 0.93 for multi-item scales in the original English version of the QLQ-BRECON23 [9]. In this study, Cronbach’s alpha is reported in the Results section.

*QLQ-BR23*: The QLQ-BR23, developed by the EORTC, is a breast-specific module to assess body image, sexual functioning, sexual enjoyment, future perspective, systemic therapy side effects, breast symptoms, arm symptoms, and upset by hair loss [7]; it comprises 23 items, and scoring was performed according to the EORTC scoring manual [13].

### 2.5. Data Analysis

Descriptive statistics were used to analyze the general and clinical characteristics of the participants. In item analysis, item–total correlation was performed to ensure that its values were at least 0.30 [14]. The distribution of normality was determined by kurtosis ranging from −3.0 to 3.0. The value of Cronbach’s alpha after each item was deleted was also considered for item selection in the item analysis process. For multiple items, the reliability of each scale was assessed using Cronbach’s alpha. To ensure the construct validity for multiple items, CFA using structural equation modeling (SEM) was performed using SPSS AMOS ver. 21 (SPSS Inc., Chicago, IL, USA). Model indices, namely chi-square goodness-of-fit (χ^2^/df), normed fit index (NFI), root mean square error of approximation (RMSEA), standardized root mean square residual (RMR), and comparative fit index (CFI), were calculated to estimate the goodness of model fit. Values indicating an acceptable fit were NFI ≥ 0.90, CFI ≥ 0.90, standardized RMR ≤ 0.08, and RMSEA < 0.08 [12].

Known-group validity was assessed by calculating the effect size (ES) of single questions according to clinical characteristics such as cancer stage (stage 0–II versus III–IV), type of BR (implant-based versus flap-based), and nipple preservation (yes versus no). The criteria for the known-group were based on a prior international validation study by the EORTC QoL group [9]. In the present study, the following rule of thumb for the ES (*d*) was applied: very small (0.01), small (0.2), medium (0.5), large (0.8), very large (1.2), and huge (2.0) [15]. Pearson’s correlation coefficient analysis was calculated for the relationship between QLQ-BRECON23 and QLQ-BR23 to determine the concurrent validity.

Missing data were imputed to the item mean according to the instructions in the EORTC QLQ-C30 scoring manual [15], as more than half of the items were completed. For example, conditional items (such as items 67, 68, and 72–77 in QLQ-BRECON23) were calculated as for missing data. A significance level of 0.05 was applied in this study.

### 2.6. Ethical Considerations

The Institutional Review Board (IRB) of C-National University Hospital approved this study protocol (2019-12-078).

## 3. Results

### 3.1. Participants

Table 1 and Table 2 present the general and clinical characteristics of patients. The mean age was 52.4 (±9.01) years, and 20.3% were single in terms of marital status. About 47.3% of the patients perceived being economically stable, and 20.9% unstable. Most (75%) of the patients were within one year of surgery, and 82.4% had implant and acellular dermal matrix (ADM) as the type of reconstruction. While 46 (32.4%) had nipple removal, only 2 (4.2%) had it reconstructed. Sentinel lymph node biopsy (SLNB) was the most common type of axillary dissection with 68.2%, followed by Levels 2 to 3 (axillary lymph node dissection) with 29.1%, and Level 1 (non-SLNB sampling) with 2.7%.

### 3.2. Construct Validity: Item Analysis

In item analysis, the values of item–total correlations for the QLQ-BRECON23 ranged from 0.221 to 0.633. Among the 23 items, 5 (Q71, Q72, Q73, Q74, Q75) were below 0.3 (Table 2). The kurtosis values for Q71, Q72, Q73, and Q74 were higher than 3.0. Cronbach’s alpha values for deleted items ranged from 0.803 to 0.844 (Table 2).

### 3.3. Construct Validity: CFA, Known-Group Validity

Figure 1 shows the results of the CFA. The standardized regression weights were higher than 4.0 for all multi-items. As shown in Table 3, the model had a good fit. All the goodness-of-fit indices except NFI were acceptable: chi-square goodness-of-fit (χ^2^/df) 1.737, NFI 0.804, standardized RMR 0.067, CFI 0.904, and RMSEA 0.070.

Known-group validity was assessed by comparing the single-item measures (SDS-Q74, NL-Q75, NP-Q76) between groups according to clinical features, namely stage (stage 0–II versus III–IV), type of BR (implant-based versus flap-based), and nipple preservation (yes versus no). As shown in Table 4, the ES (*d*) of NL (Q75) was 0.46 for stage. That is, the mean score of NL (Q75) in the stage 0–II group was significantly higher than that in the stage III–IV group (95% confidence interval [CI] 0.02–0.69). However, the ES for the other clinical features was very small to small, ranging from 0.01 to 0.19.

### 3.4. Concurrent Validity

To examine the concurrent validity, the correlation between the Korean QLQ-BR23 and QLQ-BRECON23 was examined. As shown in Table 4, the QLQ-BRECON23 function scale was positively correlated with the QLQ-BR23 function scale (r = 0.250, *p* = 0.013) and negatively correlated with the QLQ-BR23 symptom scale (r = −0.298, *p* < 0.001). The QLQ-BRECON23 symptom scale was negatively correlated with the QLQ-BR23 function scale (r = −0.271, *p* = 0.001) and positively correlated with the QLQ-BR23 symptom scale (r = 0.494, *p* < 0.001) (Table 5).

### 3.5. Reliability

Cronbach’s alpha for TS and DS on the QLQ-BRECON23 function scale were 0.68 and 0.61, respectively. Cronbach’s alpha for SX, SBC, SNC, and SSU on the QLQ-BRECON23 symptom scale were 0.87, 0.86, 0.68, and 0.70, respectively.

## 4. Discussion

The Korean version of the QLQ-BRECON23 was found to be valid and reliable in this study with female cancer patients who underwent BR. Considering the recent increase in BR in breast cancer in Korea, these patients’ specific QoL can be assessed using the Korean version of the QLQ-BRECON23. Before the QLQ-BRECON23 was developed, a core outcome set for BR was measured using the EORTC QLQ-BR23 [7]. However, women who experience BR are more likely to focus on their changed body shape or function, such as nipples or scars, according to the types of surgery methods. These detailed attributes of QoL have been reflected in the QLQ-BRECON23 for women before and after BR [9]. In this study, the Korean version of the QLQ-BRECON23, a tool that reflects these attributes, was found to be suitable for measuring the QoL of Korean women who underwent BR. Through construct validity for multiple items using CFA, we confirmed that the six-scale factor structure model was appropriate and that the scale had a structure similar to that of the original English version QLQ-BRECON23.

Similar to a previous study [9], the reliability of the QLQ-BRECON23 symptom scale was adequate, ranging from 0.68 to 0.87, but the value of Cronbach’s alpha on the function scale was lower than that on the symptom scale. To confirm the concurrent validity, we calculated the correlation coefficient between the QLQ-BRECON23 and QLQ-BR23 scales. As a result, the higher the QLQ-BRECON23 function score, the higher the QLQ-BR23 function score, and the lower the QLQ-BR23 symptom score. That is, the two measures are presumably related constructs.

As for the known-group validity, similar to the study by Winter et al. [9], this study found that the responses to the three single items, namely satisfaction with donor site, loss of nipple, and preservation of nipple, varied according to the patients’ clinical features. That is, the three single-item scores were higher in the lower cancer stage, flap-based, and nipple-preserved/reconstructed groups. For example, the score for the item “Has the loss of your nipple been a problem to you?” (Q75) was higher in the stage 0–II group than in the stage III–IV group, and its ES was moderate in this study. The items “Satisfaction with donor scars” (Q74) and “Has the preservation or reconstruction of your nipple helped you come to terms with the disease or treatment?” (Q76) showed higher scores in the nipple-preserved group than in the without-nipple group, and the ES was small in this study.

The causes for these results can be inferred from the sample characteristics. In connection with known-group validity, the value of item-total correlations in item analysis for donor-site-related items such as DS and SDS tended to be low, and a floor effect was observed. The reason can be inferred to be sample bias. In South Korea, the main type of BR is implant-based [16], and, in this study, more than 80% of the participants had an implant-type BR. As mentioned earlier, in South Korea, BR surgery has been covered by the NHIS since 2015, which is a universal and compulsory system [16]. That is, women with breast cancer generally enrolled NHIS, and healthcare providers are required to submit claims for reimbursement for BR surgery. The NHIS offers a cost-effective fee and reviews the suitability of BR in South Korea. Several previous studies [17,18] have reported the cost-effectiveness analysis of BR options from various perspectives, and cost-effective implant-based reconstruction has been recommended in South Korea. Therefore, implant surgery is on the rise, while flaps and donners are on the decline. For this reason, the mean score of Q71–74, which are items affected by the Korean medical system, was skewed and floored.

Moreover, some participants reacted sensitively to sexuality-related items (Q56–Q59), namely “Have you been feeling less sexually attractive as a result of your disease or treatment?” “Have you felt uncomfortable in intimate situations?” “Has the role of your breast in your sexuality been affected by your disease or treatment?” and “Has any loss of pleasurable sensations of your breast been a problem to you?” Therefore, it is necessary to secure anonymity and individual privacy when using the paper-pencil method in field surveys.

Regarding the usefulness of the QLQ-BRECON23, according to the readability survey, participants completed the questionnaire (23 items) within 15–20 min. In the items, there were no confusing words or hard-to-understand terms. For a thorough translation-back translation process with the EORTC group, we minimized the confusion of the words.

Several limitations of this study should be considered when applying the Korean version of the QLQ-BRECON23. First, as described above, the sample was biased in terms of types of BR. Moreover, we recruited participants from a single breast cancer center in a university hospital. Second, compared with the QLQ-BRECON23 development study [9], the research conditions were somewhat different in terms of sample size and sampling method (prospective cohort sample). Thus, we could not compare the timing of reconstruction (delayed versus immediate reconstruction) of QLQ-BRECON23. Nevertheless, it was concluded that the Korean version of the QLQ-BRECON23 is valid and reliable for capturing QoL in women undergoing BR. Based on this validation research, prospective/retrospective QoL change and its related factors can be explored in further studies with the Korean population.

## 5. Conclusions

The construct validity of the multi-item scales in the QLQ-BRECON23 using CFA was appropriate, and known-group validity for the single-item measures showed a small-to-medium effect size. The Korean version of the QLQ-BRECON23 was significantly correlated with the QLQ-BR23, and the reliability of the symptoms and function scales was acceptable. Therefore, the Korean version of the QLQ-BRECON23 is a valid and reliable tool to measure QoL in women undergoing BR. Based on this study, BR-focused QoL can be assessed in Korean women undergoing BR using the QLQ-BRECON23.

## Figures and Tables

**Figure 1 ijerph-17-09163-f001:**
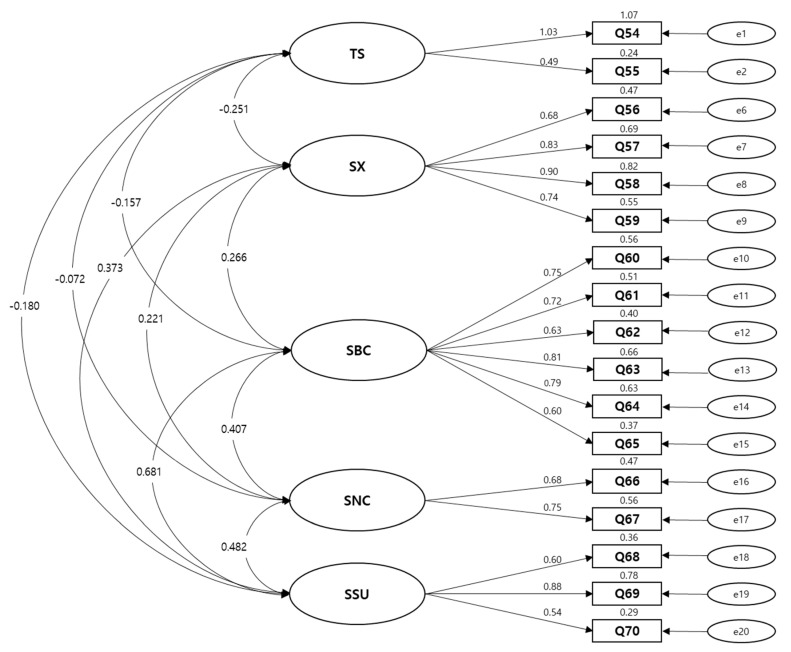
Factor structure of multi-item of Korean version of BRECON23. TS: Treatment side effect, SX: Sexual function with breast, SBC: Satisfaction with breast cosmetic, SNC: Satisfaction with nipple cosmetic, SSU: Satisfaction with surgery.

**Table 1 ijerph-17-09163-t001:** Characteristics of participants (*n* = 148).

Characteristics	Category	*n* (%) or Mean (±SD)
Age, years	Range: 33–81	52.41 (±9.01)
Religion	Yes	71 (48)
	No	77 (52)
Education level	Primary school	10 (6.8)
	Middle school	13 (8.8)
	High school	56 (37.8)
	Above college	69 (46.6)
Marital status	Single	30 (20.3)
	Married	118 (79.7)
Perceived economical	Unstable	31 (20.9)
status	Fair	47 (31.7)
	Stable	70 (47.3)
Occupation	Employed	48 (32.4)
	Unemployed	100 (67.6)
Perceived health status	Range: 1–4	3.09 (±0.88)
Presence of Comorbidity ^a^	Hypertension	25 (16.9)
	Diabetes Mellitus	12 (8.1)
	Thyroid disease	9 (6.1)
	Kidney disease	2 (1.4)
	Gastrointestinal diseases	3 (2.0)
	Cancer	2 (1.4)
	Others	7 (7.4)
ECOG PS ^b^	0	28 (18.9)
	1	97 (65.5)
	2	22 (14.9)
	3	1 (0.7)

^a^ Presence of Comorbidity: multiple response. ^b^ ECOG PS: Eastern Cooperative Oncology Group Performance Status.

**Table 2 ijerph-17-09163-t002:** Clinical characteristics of participants (*n* = 148).

Clinical Characteristics	Categories	*n* (%)
Years after surgery	≤1 year	111 (75,0)
	>1 year	37 (25.0)
Type of breast reconstruction	Implant alone	11 (7.4)
	Implant and ADM ^a^	122 (82.4)
	LD ^b^ flap and implant	1 (0.7)
	Autologous LD flap	2 (1.4)
	TRAM ^c^ pedicle/free flap	3 (2.0)
	Delayed	9 (6.1)
Type of axillary dissection	SLNB ^d^	101 (68.2)
	Level 1 (non-SLNB sampling)	4 (2.7)
	Levels 2~3 (axillary lymph node dissection)	43 (29.1)
Nipple preservation	Yes	100 (67.6)
	No	48 (32.4)
Nipple reconstruction	Yes	2 (4.2)
	No	46 (95.8)
Stage	Stage 0	16 (10.8)
	Stage Ⅰ	67 (45.3)
	Stage Ⅱ	50 (33.8)
	Stage Ⅲ	14 (9.5)
	Stage Ⅳ	1 (0.7)
Tumor type	DCIS (ductal carcinoma in situ)	21 (14.2)
	Invasive	1 (0.7)
	DCIS (ductal carcinoma in situ) and invasive	114 (77.0)
	Phyllodes tumor	1 (0.7)
	Others	11 (7.4)
Lymph node invasion	Negative	105 (70.9)
	Positive	43 (29.1)
Postoperative treatment	Radiotherapy (RTx)	2 (1.4)
	Chemotherapy (CTx)	70 (47.3)
	Hormone therapy (Hormone Tx)	25 (16.9)
	No treatment	32 (21.6)
	RTx & CTx	9 (6.1)
	RTx & Hormone Tx	1 (0.7)
	CTx & Hormone Tx	4 (2.7)
	RTx & CTx & Hormone Tx	5 (3.4)
Radiotherapy	Chest wall	15 (88.2)
	Chest wall & Axilla	2 (11.8)

^a^ ADM: Acellular Dermal Matrix. ^b^ LD: Latissimus Dorsi. ^c^ TRAM: Transverse Rectus Abdominus. ^d^ SLNB: Sentinel Lymph Node Biopsy.

**Table 3 ijerph-17-09163-t003:** Item analysis.

Scales	Subscale	Items (Range: 1–4)	Mean ± SD	Item–Total Correlations	Kurtosis	If Deleted Item Cronbach’s Alpha
Symptom scales	TS	Q 54	2.04 ± 0.92	0.396	−0.559	0.844
		Q 55	1.59 ± 0.86	0.365	0.805	0.841
	DS	Q 71	1.71 ± 0.19	0.264	31.223	0.825
		Q 72	1.71 ± 0.22	0.239	76.631	0.824
		Q 73	2.57 ± 0.26	0.221	30.965	0.826
	NL	Q 75	2.76 ± 0.66	0.221	2.241	0.826
Function scales	SX	Q 56 *	3.25 ± 0.89	0.365	−0.382	0.819
		Q 57 *	3.49 ± 0.80	0.335	0.573	0.821
		Q 58 *	3.37 ± 0.84	0.391	−0.282	0.818
		Q 59 *	3.41 ± 0.85	0.477	0.567	0.813
	SBC	Q 60	2.43 ± 0.98	0.516	−0.877	0.810
		Q 61	2.31 ± 0.97	0.621	−0.583	0.804
		Q 62	2.33 ± 1.02	0.626	−1.063	0.803
		Q 63	2.06 ± 0.94	0.621	−0.569	0.804
		Q 64	2.38 ± 0.95	0.601	−0.749	0.805
		Q 65	2.18 ± 0.97	0.542	−0.763	0.809
	SNC	Q 66	2.67 ± 0.99	0.377	−0.066	0.819
		Q 67	2.03 ± 0.94	0.448	0.681	0.814
	SSU	Q 68	1.84 ± 0.85	0.420	0.068	0.816
		Q 69	2.56 ± 0.87	0.633	−0.764	0.804
		Q 70	2.86 ± 0.91	0.395	−0.904	0.819
	SDS	Q 74	1.43 ± 0.16	0.225	68.599	0.825
	NP	Q 76	3.13 ± 0.77	0.381	0.200	0.817

* revered items. TS: Treatment side effect, DS: Donor site symptoms, NL: Loss of nipple, SX: Sexual function with breast SBC: Satisfaction with breast cosmetic, SNC: Satisfaction with nipple cosmetic, SSU: Satisfaction with surgery, SDS: Satisfaction with donor scars, NP: Preserve/reconstruct nipple.

**Table 4 ijerph-17-09163-t004:** Known-group validity in single items.

Clinical Characteristics	Categories	SDS (Q74)	NL (Q75)	NP (Q76)
		Mean ± SD	Mean ± SD	Mean ± SD
Stage	Stage 0–II	1.43 ± 0.16	2.79 ± 0.60	3.11 ± 0.79
	Stage III–IV	1.41 ± 0.15	2.44 ± 0.94	3.25 ± 0.63
	ES (*d*)	0.13	0.46	0.18
	95% CI	−0.08–0.08	0.02–0.69	−0.49–0.23
Type of breast reconstruction	Implant-based	1.40 ± 0.80	2.78 ± 0.64	3.11 ± 0.78
	Flap-based	1.52 ± 0.83	2.80 ± 0.09	3.18 ± 0.74
	ES (*d*)	0.15	0.04	0.10
	95% CI	−0.20–0.06	−0.54–0.51	−0.73–0.58
Nipple preservation	Yes	1.43 ± 0.18	2.76 ± 0.02	3.16 ± 0.90
	No	1.41 ± 0.08	2.75 ± 1.16	3.03 ± 0.37
	ES (*d*)	0.18	0.01	0.19
	95% CI	−0.02–0.07	−0.07–0.34	−0.07–0.34

SDS (Q74): Satisfaction with donor scars, NL (Q75): Loss of nipple, NP (Q76): Preserve/reconstruct nipple, ES: effect size, 95% CI: 95% Confidence Interval.

**Table 5 ijerph-17-09163-t005:** Correlations between BRECON23 and BR23.

r-Values	BRECON23 Functions	BRECON23 Symptoms
BR23 Functions	0.250 (*p* = 0.013)	−0.271 (*p* = 0.001)
BR23 Symptoms	−0.298 (*p* < 0.001)	0.494 (*p* <0.001)

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
