# Peer review of "Korean Translation and Psychometric Evaluation of Korean Version EORTC QLQ-BRECON23"

_ijerph, 2020, doi:10.3390/ijerph17249163_

Round 1
Reviewer 1 Report
This manuscript describes the psychometric properties of the Korean translated version of the QLQ-BRECON23 questionnaire, together with the translation process. The manuscript is well written and the described topic is scientifically important; the paper will be a reference work for subsequent Korean studies investigating the quality of life after breast reconstruction surgery in breast cancer patients. The following minor points are recommended to be fixed before approval:
- The manuscript title suggests that the Korean translation was accompanied by cultural adaptation - however, according to line 92, there were no questions or revisions for the preliminary translation of the Korean version, so cultural adaptation was apparently not necessary / not approached (although the Authors noticed that some participants reacted sensitively to sexuality-related items - Line 249). Accordingly, the title is suggested to be amended, not emphasizing cultural adaptation but indicating that the psychometric characteristics of the translated questionnaire were analysed (this is currently not mentioned in the title).
- Lines 35-36: "The rate of occurrence of breast cancer in South Korea is 12.7% for those in their 30s and 37.1% for those in their 40s, which is lower than in Western countries [5]". Please adjust the wording to make it more clear that these rates refer to the age distribution of breast cancer cases, not to incidence rates in various age groups.
- Line 78: inability to understand or complete the questionnaire was an exclusion criteria. Please report the number of patients excluded based on this criteria. In case several patients were not eligible for this reason, explanatory patient heterogeneity factors (education level? urban vs rural? other cultural factors?) might also be explored and discussed.
- The translated Korean language questionnaire could be attached to the paper as appendix, if allowed by existing intellectual property rights / copyright agreements.
Author Response
Review 1
Comment#1: The manuscript title suggests that the Korean translation was accompanied by cultural adaptation - however, according to line 92, there were no questions or revisions for the preliminary translation of the Korean version, so cultural adaptation was apparently not necessary / not approached (although the Authors noticed that some participants reacted sensitively to sexuality-related items - Line 249). Accordingly, the title is suggested to be amended, not emphasizing cultural adaptation but indicating that the psychometric characteristics of the translated questionnaire were analysed (this is currently not mentioned in the title).
=> Response: The title was changed as reviewer’s suggestion.
“Korean translation and psychometric evaluation of the Korean version EORTC QLQ-BRECON23”
Comment#2: Lines 35-36: "The rate of occurrence of breast cancer in South Korea is 12.7% for those in their 30s and 37.1% for those in their 40s, which is lower than in Western countries [5]". Please adjust the wording to make it more clear that these rates refer to the age distribution of breast cancer cases, not to incidence rates in various age groups.
=> Response: We changed the word of “rate of occurrence of breast cancer”
"The age distribution of breast cancer cases in South Korea is 12.7% for those in their 30s and 37.1% for those in their 40s, which is lower than in Western countries [5]".
Comment#3: Line 78: inability to understand or complete the questionnaire was an exclusion criteria. Please report the number of patients excluded based on this criteria. In case several patients were not eligible for this reason, explanatory patient heterogeneity factors (education level? urban vs rural? other cultural factors?) might also be explored and discussed.
=> Response: The number of patients excluded based on this criteria was two those could not understand Korean. We added this statement in text as below.
“For this reason, the number of ineligible patients of this reason was two international women who did not understand Korean.”
Comment#4: The translated Korean language questionnaire could be attached to the paper as appendix, if allowed by existing intellectual property rights / copyright agreements.
=> Response: The copy right of this Korean version is belong to EROTC. So we could not attach the Korean version.
Reviewer 2 Report
It was my pleasure to review the article titled Korean Translation and Cultural Adaptation of 2 Korean Version EORTC QLQ-BRECON23 by Soo-Kyung Bok et al. The article I is very interesting and the results even the limitations of the study are important to evaluate the possibility to use a quality-of-life questionnaire for women undergoing breast reconstruction, that may become a worldwide questionnaire.
I have some somme comments to the authors.
Introduction
The introduction could be re-organised explaining more clear, the incidence of Breast cancer worldwide and in Korea, the actual treatments and complications in Breast reconstruction and afterwards the aim of this study.
Introduction
Line 49 Try to rephrase this sentence.
To overcome this limitation, the EORTC 49 developed the QLQ-BRECON23 for measuring QoL in women undergoing BR [8], and the validity 50 and reliability of the original English version were confirmed through an international validation 51 process including seven countries [9].
3.2. Construct validity: item analysis
In figure 1 caption
SDS: Satisfaction with donor scars is not represented in the graph
Discussion
I agree with the authors about the limitations of the study and it will be important to continue with the study to reduce these limitations.
Author Response
Review 2
Comment#1: Introduction The introduction could be re-organised explaining more clear, the incidence of Breast cancer worldwide and in Korea, the actual treatments and complications in Breast reconstruction and afterwards the aim of this study.
Response: The first paragraph and the second paragraph was exchanged. Along with changing those sentences, the reference number was also reorganized.
Comment#2: Introduction Line 49 Try to rephrase this sentence.
To overcome this limitation, the EORTC 49 developed the QLQ-BRECON23 for measuring QoL in women undergoing BR [8], and the validity 50 and reliability of the original English version were confirmed through an international validation 51 process including seven countries [9].
Response: We changed this sentence.
Comment#3: 3.2. Construct validity: item analysis In figure 1 caption :SDS: Satisfaction with donor scars is not represented in the graph
Response: The single item such as NL, SDS, NP could not perform the CFA. The single item was evaluated using Known-group validity. This information was presented in statistical analysis (page 8).
“Known-group validity was assessed by calculating the effect size (ES) of single questions according to clinical characteristics such as cancer stage (stage 0–II versus III–IV), type of BR (implant-based versus flap-based), and nipple preservation (yes versus no)”
This manuscript is a resubmission of an earlier submission. The following is a list of the peer review reports and author responses from that submission.